# The Impact of an Integrated Program of Return-to-Field and Targeted Trap-Neuter-Return on Feline Intake and Euthanasia at a Municipal Animal Shelter

**DOI:** 10.3390/ani8040055

**Published:** 2018-04-13

**Authors:** Daniel D. Spehar, Peter J. Wolf

**Affiliations:** 1Independent Researcher, 4758 Ridge Road, #409, Cleveland, OH 44144, USA; danspehar9@gmail.com; 2Best Friends Animal Society, 5001 Angel Canyon Road, Kanab, UT 84741, USA

**Keywords:** feral cats, return-to-field (RTF), trap-neuter-return (TNR), targeted TNR, municipal animal shelter, feline intake, feline euthanasia, live release rate (LRR), community cat program (CCP)

## Abstract

**Simple Summary:**

Dramatic declines in the number of cats admitted to and euthanized at U.S. shelters have taken place in recent decades. Still, millions of cats, many of them free-roaming, enter shelters each year. At some facilities, as many as 70% of feline admissions are euthanized, and it is estimated that, nationally, up to one million or more cats are euthanized each year. New approaches, including return-to-field (RTF) and targeted trap-neuter-return (TNR) appear to have transformative potential. The present study examines changes in feline intake and euthanasia, as well as impacts on associated metrics, at a municipal animal shelter in Albuquerque, New Mexico, after formal RTF and targeted TNR protocols, collectively referred to as a community cat program (CCP), were added to ongoing community-based TNR efforts and a pilot RTF initiative. As part of the three-year CCP, 11,746 cats were trapped, sterilized, vaccinated and returned or adopted. Feline euthanasia at the Albuquerque Animal Welfare Department (AAWD) declined by 84.1% and feline intake dropped by 37.6%; the live release rate (LRR) increased by 47.7% due primarily to these reductions in both intake and euthanasia. Modest increases in the percentage of cats returned to owner (RTO) and the adoption rate were also observed, although both metrics decreased on an absolute basis, while the number of calls to the city about dead cats declined.

**Abstract:**

Available evidence indicates that overall levels of feline intake and euthanasia at U.S. shelters have significantly declined in recent decades. Nevertheless, millions of cats, many of them free-roaming, continue to be admitted to shelters each year. In some locations, as many as 70% of cats, perhaps up to one million or more per year nationally, are euthanized. New approaches, including return-to-field (RTF) and targeted trap-neuter-return (TNR) appear to have transformative potential. The purpose of the present study was to examine changes in feline intake and euthanasia, as well as additional associated metrics, at a municipal animal shelter in Albuquerque, New Mexico, after institutionalized RTF and targeted TNR protocols, together referred to as a community cat program (CCP), were added to ongoing community-based TNR efforts and a pilot RTF initiative. Over the course of the CCP, which ran from April 2012 to March 2015, 11,746 cats were trapped, sterilized, vaccinated, and returned or adopted. Feline euthanasia at the Albuquerque Animal Welfare Department (AAWD) declined by 84.1% and feline intake dropped by 37.6% over three years; the live release rate (LRR) increased by 47.7% due primarily to these reductions in both intake and euthanasia. Modest increases in the percentage of cats returned to owner (RTO) and the adoption rate were also observed, although both metrics decreased on an absolute basis, while the number of calls to the city about dead cats declined.

## 1. Introduction

### 1.1. Feline Intake and Euthanasia Trends at U.S. Animal Shelters: A Historical Perspective

Although historical data are scarce, available evidence indicates that the number of cats currently admitted to animal shelters in the United States is only a small fraction, perhaps less than 10%, of what it was at its peak in the 1930s [1,2,3]. Moreover, a precipitous drop in the number of cats euthanized in shelters has occurred since the initiation of widespread sterilization efforts targeted at pet cats (and dogs) beginning in the 1970s, when more than 90% of animals entering shelters were euthanized [1,3]. Despite these improvements, it is estimated that as many as 70% of cats entering some U.S. shelters continue to be euthanized [4,5,6,7,8].

In general, free-roaming cats remain the greatest source of feline intake at U.S. shelters [9,10] and cats deemed “feral”, and thus unadoptable, are typically euthanized [11,12]. Expanded use of colony-level trap-neuter-return (TNR) programs and the initiation of low-cost sterilization campaigns aimed at pet, as well as stray cats, in underserved communities have been associated with continued reductions in feline intake at shelters since the mid-1990s [2,13]; however, to date, these efforts alone have been insufficient to reshape the shelter paradigm on a broad scale. Studies of shelters in several states (Colorado, New Hampshire, North Carolina, and Ohio) indicate varying trends in feline intake and euthanasia over the past two decades [13,14,15,16]. It is estimated that between 860,000 and 1.4 million cats continue to be euthanized annually in U.S. shelters [8,17]. However, new approaches developed over the past decade appear to have transformative potential. Intensive TNR programs have significantly reduced feline shelter intake from targeted areas [10,18], and return-to-field (RTF) programs (a.k.a., Feral Freedom or shelter-neuter-return) have reduced the euthanasia of cats at municipal shelters to levels once thought to be unattainable [5,7,18]. 

The present study examines changes in feline intake and euthanasia, as well as additional associated metrics, at a municipal animal shelter in Albuquerque, New Mexico, after institutionalized RTF and targeted TNR protocols, together referred to as a community cat program (CCP), were added to ongoing community-based TNR efforts and a pilot RTF initiative.

### 1.2. Site Description and Lead-Up to the Initiation of the Community Cat Program

Albuquerque is located in the high desert of north-central New Mexico and straddles the Rio Grande River [19]. It has an estimated population of 559,277 [20] and serves as the seat of Bernalillo County, which has a total estimated population of 676,953 [21]. The Albuquerque Animal Welfare Department (AAWD) operates two open-admission shelters (facilities that generally accept any animal in need, including those with little chance of being rehomed due to issues of age, health, or temperament [22]) that serve the city of Albuquerque [23] and, by contract, the remainder of Bernalillo County [24]. Bernalillo County is scheduled to open its own animal shelter in 2018 [24]. Historically, companion animals were required to be held for a minimum of five days prior to disposition if admitted to AAWD with identification, or for three days if admitted without it. Cats deemed “feral,” despite nearly always lacking any form of identification (Albuquerque began requiring the microchipping of pets in 2006 [25]), were subjected to the longer holding time until 2010 [24]. After the mandatory holding time, cats deemed unadoptable due to health or temperament were euthanized. Several thousand free-roaming cats were typically brought to AAWD shelters each year by residents, animal control officers, and commercial trappers [24,26,27]. The heavy inflow of free-roaming cats into Albuquerque’s municipal shelters made the euthanasia of cats to clear space for new admissions a common occurrence [24,26,28]. AAWD admitted 12,339 cats in 2007 and 7562 (61.3%) were euthanized; of those, 2737 (36.2%) were categorized as feral. In 2008, 11,495 cats were admitted to AAWD; of the 7234 (62.9%) cats euthanized, 2612 (36.1%) were categorized as feral.

In an attempt to reduce the number of cats being euthanized each year, especially the number of cats deemed feral [26,27], AAWD began collaborating with local non-profit groups in 2008 to promote the use of TNR [18,26,27]. A popular low-cost trap rental program operated by AAWD that permitted local residents to capture and bring in free-roaming cats for shelter admission (and most often euthanasia) was eliminated [18,26,28]. In its place, a community trap bank, operated by local non-profit groups, was established; this allowed residents to use traps free of charge, but only for TNR [26]. In addition, from that point forward, calls from residents to AAWD about free-roaming cats were referred to TNR groups; however, residents were still able to drop off free-roaming cats whom they had trapped on their own. Once these new shelter policies were implemented, a near-immediate reduction in feline euthanasia was observed [26].

The same year, Street Cat Companions (SCC), the TNR program of New Mexico Animal Friends (NMAF), partnered with Animal Humane New Mexico (AHNM), a private shelter in Albuquerque, and AAWD to offer free spay-neuter surgeries for feral cats [18,26,27]. To do this, AAWD began to cover the customary $15 co-payment previously paid by residents and rescue groups for each cat sterilized (the remainder being paid by SCC and AHNM, using funds obtained through various grants). The agency’s goal was generally to promote the practice of TNR among residents and to stimulate large-scale community trapping efforts [26]. Due in part to the availability of free sterilization surgeries, SCC was able to target many of the large free-roaming cat colonies in Bernalillo County and achieve high sterilization rates within those colonies [18,26]. Such colonies were selected for targeting based upon the volume of calls received by SCC from residents about free-roaming cats, and colony locations were informally tracked using Google Maps [29].

In fewer than three years after initiation of the aforementioned programs (year-end 2010), feline euthanasia at the AAWD had fallen 31.9% from 2007 levels (from 7562 to 5147), while intake was reduced by 21.2% (from 12,339 to 9717). The euthanasia rate for cats declined from 61.3% in 2007 to 53.0% in 2010. Moreover, the number of cats euthanized because they were categorized as feral fell from 2737 to 1196 (23.2% of all cats euthanized).

Despite the reductions in feline intake and euthanasia associated with the foregoing initiatives, shelter leadership recognized that additional changes in shelter protocol were needed to build upon these trends, and to effectuate an end to the routine euthanasia of healthy feral cats [26]. Consequently, in 2011, AAWD began, on a trial basis, a RTF program whereby all suitably healthy free-roaming cats surrendered to the agency were transferred to SCC or other rescue groups after examination, sterilization, and vaccination; these cats were then returned to locations of capture rather than being admitted to the shelter and likely euthanized [18,26,27]. An estimated 1000 cats were handled in this fashion during the program’s first year [26]. By the end of 2011, annual feline euthanasia at AAWD had dropped to 3503 and the rate of feline euthanasia was down to 35.7% of intake (9810 cats); the euthanasia of cats designated as feral had been reduced to 473 (13.5% of all cats euthanized). Less than two months later (approximately 3.5 years after the agency’s trap rental program was discontinued and TNR became the preferred method of managing free-roaming cats), in February 2012, AAWD euthanized a cat for the last time solely because the cat was labelled as feral [26,27]. 

AAWD received guidance from Best Friends Animal Society (BFAS), which three years earlier had provided funding for the first large-scale RTF program in the U.S., in Jacksonville, Florida [18], as it implemented its trial RTF program in 2011. In 2012, AAWD was enrolled in a pilot community cat program (CCP) originated and sponsored by BFAS, in collaboration with PetSmart Charities, Inc. (PCI), for the purpose of integrating a RTF program with targeted TNR at a municipal animal shelter.

## 2. Materials and Methods

### 2.1. The Albuquerque Animal Welfare Department’s Community Cat Program: Integrating the Use of Return-to-Field and Targeted Trap-Neuter-Return (TNR)

The CCP was modelled after the Feral Freedom program established in Jacksonville, where feline euthanasia was reduced by 92% over six years [18]. However, in Albuquerque a targeted TNR component was paired with the RTF initiative from the onset, rather than added three years after inception as in Jacksonville [18]. The CCP was set up as a three-year program, commencing in April 2012 and concluding in March 2015, with a goal of sterilizing 10,500 cats [30]. 

The RTF portion of the CCP was structured so that all healthy free-roaming cats entering the shelter were transferred to BFAS staff (two full-time employees were hired to work in conjunction with AAWD to operate the program), who arranged for the cats to be sterilized at a high-quality, high-volume spay-neuter clinic operated by AHNM. BFAS personnel then returned the cats to the locations where they had been trapped [18,31]. In addition to being sterilized, before being returned to the field, cats were ear-tipped and received rabies and rhinotracheitis/calciviris/panleukopenia (FVRCP) vaccinations, as well as flea treatment and an antibiotic injection (cefovecin sodium, a.k.a., Convenia), as appropriate [31]. Original protocol called for all free-roaming cats without serious illness or injury to be returned to locations of capture after recovery from sterilization surgery; however, over time, as feline intake declined and more shelter space became available, some sociable cats were admitted for adoption or transferred to rescue groups [31,32].

Targeted TNR was performed in areas of the city determined to be sources of high feline intake. AAWD periodically provided BFAS with a list of “hot spots” from which a disproportionate number of free-roaming cats were arriving at city shelters [31]. BFAS staff, with the assistance of volunteers, trapped cats at selected locations and delivered them to AHNM for examination, sterilization, and vaccination. After recovery, BFAS personnel returned the cats to their locations of capture. Targeted trapping was also performed at release sites of cats returned as part of the RTF program [18,31,32]. Such sites were selected based upon the theory, termed the “red-flag cat model” (RFCM), that locations within a community capable of sufficiently supporting one cat were likely home to additional unsterilized cats [18,30]. Thus, the initial cat trapped and returned to a new location acted as an indicator, or red flag, alerting program staff to the potential presence of other cats. Information about the CCP was disseminated via concentrated community outreach, including door-to-door canvassing (a.k.a., block walking), the distribution of door hangers, and the hosting of educational events, in the neighborhoods where targeted TNR took place [33].

Cats were not relocated unless their home environments were deemed too dangerous for return—a situation that occurred only rarely. Based upon the results in Jacksonville, microchipping of cats was not part of the Albuquerque CCP protocol. Initially, cats returned to the field in Jacksonville were microchipped in order to track the number who were impounded more than once or became victims of nefarious human behavior; the practice was discontinued when “no evidence of mistreatment of returned cats turned up” [18] (p. 85). Zip codes where cats were trapped and returned included those within the city of Albuquerque as well as some in smaller towns and unincorporated areas of Bernalillo County [18].

### 2.2. Data Collection

For the purposes of the present study, all data pertaining to the CCP and AAWD results from the years 2011 to 2016 were obtained from BFAS; AAWD results prior to 2011 were acquired directly from AAWD. 

CCP cat and surgery information was entered by CCP staff and volunteers into an internally-built database created by BFAS. The data was updated several times per month throughout the course of the program. Statistics were assessed monthly to evaluate the progress of the program toward overall sterilization surgery goals [34]. AAWD employed Chameleon sheltering software; information was inputted by shelter staff and available for tracking on a daily basis [35].

AAWD shelter data tracked as part of the CCP included live intakes, live outcomes, and other outcomes, including euthanasia. Intake and euthanasia statistics were recorded by age: adult and kitten (≤5 months of age); admissions of kittens two months of age and under were also tracked. All sterilization surgeries, whether performed as part of the RTF or targeted TNR programs, as well as the number of cats returned to colony sites, were documented. Outcomes for cats returned to colony sites were not specifically tracked as part of the CCP [34].

### 2.3. Data Analysis

Annual feline intake and euthanasia data (broken down by age, where possible) for each of the CCP years were compared to AAWD data from years before and after the CCP, as well as to comparable data from other communities where similar programs have been implemented. In addition, the number of cats included in the RTF component was compared to those included in the TNR component for each year of the CCP, with a particular emphasis on instances of RTF and TNR cats originating from the same location. Due to the small sample size involved (e.g., 3 program years), no statistical analysis was attempted.

## 3. Results

A total of 11,746 cats (8160 (69.5%) adults and 3586 (30.5%) kittens) were enrolled in the three-year Albuquerque CCP. Sterilization surgery was performed on a total of 11,038 cats, 8851 (80.2%) as part of the targeted TNR program and 2187 (19.8%) under the RTF initiative. The combined number of sterilization surgeries fluctuated over the course of the CCP: Year 1 (4/12 to 3/13): 3723; Year 2 (4/13 to 3/14): 3981; Year 3 (4/14 to 3/15): 3334. Sterilizations associated with the RTF program declined each year as a percentage of the total number performed: Year 1: 25.9%; Year 2: 19.1%; Year 3: 13.9% (Figure 1). Over the course of the CCP, the number of male cats sterilized slightly exceeded females 5589 (50.6%) to 5449 (49.4%) and more adults were sterilized than kittens, 7551 (68.4%) to 3487 (31.6%). A total of 642 cats (5.5%) were discovered after program enrollment to have been previously sterilized.

In total, 10,738 cats (91.4%) were returned to colony sites as part of the CCP; 946 (8.0%) were adopted from AAWD, transferred to rescue groups for adoption, or placed in foster care; 34 (0.3%) died in surgery, pre-operatively, post-operatively, or in care; 20 (0.2%) were euthanized for serious health concerns; 6 (0.1%) were relocated because they could not be returned safely to locations of capture; and 2 (<0.1%) were returned to owner or otherwise released without undergoing surgery (Table 1). Of the cats returned to colony sites, 7654 (71.3%) were adults, 3067 (28.6%) were kittens (≤5 months of age), and the age of 17 (0.1%) was unknown. Cats originated from 1875 different colony sites across 22 zip codes. The number of cats enrolled in the CCP per zip code (as part of either the RTF or targeted TNR programs) ranged from 3 to 2032, median of 353; 69% of the cats originated from locations in 6 of the 22 zip codes in which the CCP operated (87121, 87105, 87107, 87102, 87123, and 87108).

At the end of the three-year CCP, when compared to a baseline of the 12-month period immediately prior to program inception, feline euthanasia at AAWD declined by 84.1%, from 3023 (4/2011 to 3/2012) to 480 (4/2014 to 3/2015) cats. The euthanasia rate for cats fell by 74.4% over the same period, from 30.9 to 7.9%, and feline intake was reduced by 37.6%, from 9776 to 6102. The euthanasia of kittens (≤5 months of age) declined by 89.8% from 1462 at year-end 2011 to 149 at the close of 2015 (some data were tracked only by calendar year); the euthanasia rate for kittens fell 81.8% from 32.9 to 6.0% over the same period, while kitten intake dropped by 44.4% from 4441 to 2468, overall, and 40.3% (2803 to 1672) for “newborn” kittens under two months of age (Table 2).

The live release rate (LRR), calculated using the American Society for the Prevention of Cruelty to Animals (ASPCA) formula of dividing live outcomes by intake [36], for cats at AAWD increased from 60.6% in 2011 to 89.5% in 2015; an improvement of 47.7%. Euthanasia of cats per 1000 Bernalillo County residents declined by 86.5% from 5.2 in 2011 to 0.7 in 2015; feline intake per 1000 county residents fell by 44.2% from 14.7 to 8.2. Over the same period, the number of cats reclaimed by their owners (a.k.a., returned to owner (RTO)) decreased 6.7% from 297 (3.0% of feline intake) in 2011 to 277 (5.0% of feline intake) in 2015, and the number of cats adopted from AAWD shelters decreased 21.8%, from 4264 (43.5% of feline intake) to 3333 (60.2% of feline intake). Moreover, calls from residents (via a 3-1-1 non-emergency municipal services line) about dead cats in the city of Albuquerque declined by 23.9%, from 2220 in 2011 to 1689 in 2015 (Table 2).

Results at AAWD for 2016 reflected continued declines in feline intake (5078), euthanasia (361) and euthanasia rate (7.1%), and sustained increases in LRR (91.0%) adoption rate (67.4% of feline intake, based upon 3422 adoptions), and RTO rate (5.5% of feline intake, based upon 279 cats returned to their owners); moreover, the number of 3-1-1 calls for dead cats decreased again, to 1616 (2017 results for this metric, which became available just as the present investigation was being concluded, indicate a continued decline to 1222 calls about dead cats).

## 4. Discussion

Multiple initiatives commencing in 2008, some shelter-based and others conducted by private non-profit organizations in the Albuquerque community, contributed to the significant declines in feline euthanasia and intake experienced at AAWD. The totality of these efforts can be loosely divided into four phases, a key component of which was the three-year CCP, which began in April of 2012 (Figure 2).

Phase 1 (2008 through 2010) established TNR as the preferred method of managing free-roaming cats in the city of Albuquerque. Actions included discontinuing trap rentals to residents for the purpose of bringing in feral cats, support for community-based TNR by referring calls about free-roaming cats to TNR groups, and subsidizing the cost of sterilization surgery co-payments for residents. Phase 2 (2011 through March 2012) laid the foundation for the CCP by initiating a pilot RTF program at AAWD that ended the euthanasia of cats based solely upon temperament. Phase 3 (April 2012 through March 2015), the CCP, institutionalized the use of RTF at AAWD by implementing sustainable protocols that appear to have been the primary factor in feline euthanasia dropping by 84.1%. Similarly, the integrated use of targeted TNR, including the RFCM, appear to have been the primary factor in a significant reduction (37.6%) in feline intake at AAWD as well. Phase 4 (April 2015 through 2016) consisted of the permanent adoption of RTF and targeted TNR as free-roaming cat management practices at AAWD. After the formal conclusion of the CCP, major elements of the program were retained by AAWD. Although no longer operated by BFAS (and funded by BFAS and PCI), shelter-based RTF and targeted TNR programs were continued due in large part to a municipally-funded contract between AAWD and Street Cat Hub (SCH), formerly SCC [26,32]. Under the agreement, SCH assumed the role of trapping and returning cats previously filled by BFAS. Improvements in shelter metrics attained as part of the CCP were sustained during this period.

As was observed in Jacksonville and San José, California [7,18], the addition of a RTF program to existing community and shelter-based initiatives (e.g., low-cost or free sterilization programs for pet cats and colony-level TNR) appears to have been the primary impetus behind the dramatic reductions in euthanasia at AAWD. Six years after initiation of the pilot RTF program in 2011, feline euthanasia at AAWD had declined by 93.3% (compared to the baseline year of 2010); 53% of that reduction occurred during the three-year CCP. The significant decrease (47.8%) in feline intake at AAWD over the same period is likely attributable to the addition of targeted TNR to the RTF program, as 79.2% of the observed decline in feline intake between 2010 and 2016 took place over the course of the CCP. In fact, during the pilot RTF program (2011), which included no targeting component, cat intake at AAWD increased by nearly 1% (from 9717 to 9810). Feline intake declined each year after targeted TNR began to be practiced in tandem with RTF. Similarly, after a program including targeted TNR was added to the aforementioned RTF program in Jacksonville, feline intake declined at the municipal shelter by 40.8% over four years [18,37]. A decline in the admission of stray cats at AAWD (from 6406 to 3563) comprised 60% of the total reduction in feline intake from 2011 through 2016 (the only years for which such data are available); a drop in the admission of owner-surrendered cats (from 3232 to 1345) accounted for nearly all the remainder of the overall reduction in feline intake. One factor likely contributing to the decline in owner-surrendered cats is a program initiated in 2014 (about halfway through the CPP), which provided free resources (e.g., spay or neuter surgeries, food, and advice concerning behavioral issues) to residents considering surrendering a pet [24]. Another contributing factor might have been an ongoing misclassification of cats at shelter surrender, as described by Zito et al. [38], due to an inherent inadequacy in the binary choice of admission categories (i.e., owner surrendered pet or stray), which fails to account for “semi-owned” cats receiving varying levels of direct support from humans. Hesitation on the part of caretakers to bring such cats to AAWD likely waned after the routine euthanasia of “feral” cats ceased in 2008, thereby creating an increase (of unknown magnitude) in the intake of cats presumably deemed “stray,” yet which may have been more accurately categorized as “semi-owned.”

The sharp declines noted above in feline intake and euthanasia at AAWD took place despite an estimated 2% increase in the population of Bernalillo County over the same period [39]. From 2011 to 2015, the number of cats admitted to AAWD per 1000 county residents fell by 44.2% (from 14.7 to 8.2), and the number of cats euthanized per 1000 county residents dropped by 86.5% (from 5.2 to 0.7). Elsewhere, four years after initiation of the previously referenced RTF program in San José, the number of cats admitted to the municipal shelter per 1000 county residents was reduced by 31.4% (from 10.2 to 7.0) and feline euthanasia per 1000 residents had declined by 77.8% (from 7.2 to 1.6) [7]. In Alachua County, Florida, where a two-year high-impact targeted TNR and adoption program was implemented, a 69.2% reduction in feline intake (from 13.0 to 4.0 per 1000 residents) and a 95% decrease in feline euthanasia (from 8.0 to 0.4 per 1000 residents) at the municipal shelter occurred in the targeted area (zip code 32601) versus declines of 31.3% (from 16.0 to 11.0 per 1000 residents) and 30% (from 10.0 to 7.0 per 1000 residents), respectively, in the remainder of the county [10] (Table 3). Sterilization efforts from the CCP resulted in 4.9–5.9 cats sterilized annually per 1000 residents, although additional sterilization efforts were undertaken concurrently throughout Bernalillo County (e.g., via AHNM). These rates are considerably lower than those reported in Alachua County’s targeted TNR program (57–64 cats sterilized annually per 1000 residents) [10], but similar to those from San José’s RTF program (approximately 2.7 cats sterilized annually per 1000 residents) [7]. The much greater figure resulting from targeted TNR efforts in Alachua County is most likely the result of the limited geographic area (i.e., one zip code); by contrast, the other two programs covered much larger areas and a much greater number of residents (although the CCP was still targeted in scope).

No zip code-specific feline intake data from AAWD was available to compare results from targeted and non-targeted areas. However, such data was collected by AHNM in order to track feline intake from six zip codes that were the subject of a targeted TNR program, funded by PCI, in 2011 and 2012. Results of this program indicate a two-year decline in feline intake of 62% at AHNM from the targeted zip codes compared to a reduction of only 8% from non-targeted areas [18]. These results were similar to what occurred in Alachua County [10]. Zip codes with the highest levels of stray cat and kitten intake at AHNM were selected for the targeted TNR program [40], which remained active at year-end 2016. Between 2012 and 2015, AHNM’s targeted TNR efforts resulted in the sterilization of 3489 cats in seven zip codes. Nearly 70% of the cats enrolled in the CCP originated from six of the same zip codes (Figure 3).

The RFCM appears to have played a significant role in bringing about the reductions in feline euthanasia and intake experienced at AAWD. In all, 2707 cats enrolled in the CCP (23%) originated from locations where both RTF and targeted TNR took place in a given year. Nearly 70% of these cats (1881) were trapped, sterilized, vaccinated, and returned or adopted as part of targeted TNR efforts associated with cats returned to the field from AAWD. On average, an additional 4.5 “field-origin” cats (median of 2) were enrolled in the CCP for each “shelter-origin” cat returned to field at RFCM sites; results varied by year and the range is far greater than might be suggested by the median values (Figure 4). In some cases, more than 50 “field-origin” cats were enrolled in the TNR program as a result of targeted outreach efforts in response to a single cat being brought to the shelter as a stray. This illustrates a key advantage of the RFCM, and of integrating RTF and TNR programs in general. The RFCM was part of the Albuquerque CCP from the beginning; however, in practice, it was implemented with more regularity as the number of locations in the community that were considered sources of high feline intake at AAWD were reduced over the course of the three-year program [32]. This trend is illustrated in the ratios of cats enrolled in the CCP as part of either the targeted TNR or RTF component; targeted TNR enrolled cats outnumbered RTF cats 3:1 in Year 1, 4:1 in Year 2, and >6:1 in Year 3. Similarly, RTF cats declined as a percentage of total feline intake at AAWD from 12% in Year 1 of the CCP to 7.6% in Year 3, presumably as the percentage of sterilized cats increased at enrolled colony locations. It is estimated that by the end of the CCP a 90% sterilization rate was achieved across the 1875 enrolled colonies [41].

The cats enrolled in the CCP were generally in good health, as illustrated by the low incidence of cats requiring euthanasia or dying in care, which is consistent with what has been observed at other locations where RTF [7] and targeted TNR [10] programs have been implemented. Implementation of the CCP did not alter the ratio of adult cats to kittens admitted to AAWD. More adult cats than kittens were admitted, by a ratio of 1.2:1, in both 2011 and 2015. Similarly, after initiation of a RTF program in San José, the relationship between the intake of adult cats and kittens remained stable; however, in San José more kittens than adult cats entered the shelter both before and after initiation of the program [42].

Although the percentage of feline intake at AAWD adopted into homes increased from 43.5% in 2011 to 60.2% in 2015, the number of cats adopted decreased by 21.8% from 4264 to 3333 cats. The most obvious explanation is that adoptions decreased largely as a result of reductions in feline intake. However, this was not the case in San José, where feline intake decreased nearly 21% (from 43,517 to 34,380) as a result of a four-year RTF program while adoptions increased about 1% (from 5126 to 5175) [42]. Adoptions of cats and kittens from AHNM also decreased, from 1741 cats before the CCP to 1444 cats after the CCP (although a precise comparison is not possible because AHNM data is tracked by a fiscal year that does not correspond to CCP program years or calendar years), possibly as a result of 463 fewer cats coming into AHNM. Even so, this decrease is surprising since AHNM opened a new adoption center in 2014, designed to make adoptions a more attractive and convenient option for residents. Although it is not clear from the data available why adoptions of cats and kittens from both AAWD and AHNM would have decreased over the course of the CCP, one likely factor is the reduced “supply” to both organizations as a result of the AAWD’s ongoing RTF efforts. Again, though, this trend was not observed in San José [42]. It is possible, too, that targeted TNR efforts led to an overall decrease in the number of free-roaming cats and kittens in the community (the “supply” itself). In any case, despite fewer adoptions from AAWD, the agency’s dramatic decrease in feline euthanasia resulted in the significant improvement in LRR (60.6% to 89.5%) observed over the course of the CCP.

RTO decreased from 297 in 2011 to 277 cats in 2015. This modest decline must be considered in light of the results of a national survey of U.S. households, which found that only 2% of lost cats were recovered by contacting a local shelter [43]. Viewed in this context, the increase in AAWD’s RTO as a percentage of feline intake, from 3.0% in 2011 to 5.0% in 2015, is noteworthy. By way of comparison, San José’s RTF program saw RTO decrease from 1200 cats (2.8% of feline intake) to 772 (2.2% of feline intake) over four years [42]. AAWD’s modest reduction in RTO might be seen as surprising given the prominent role RTF played in the CCP and the considerable decline in feline intake; if there are far fewer cats kept in the shelter, one might expect a sizable decrease in RTO. However, RTO decreased 6.7% while feline intake decreased 37.6%; AAWD reunited 20 fewer cats with their owners in 2015 despite 3674 fewer cats entering the shelter. One factor likely contributing to AAWD’s RTO is the mandated microchipping of pets in the city of Albuquerque making it easier for lost cats to be reunited with their owners [25,29,44]. Moreover, as has been observed elsewhere after the implementation of a RTF program [7], AAWD documented a significant decrease in the number of dead cats recorded on an annual basis.

## 5. Study Limitations

As has been noted elsewhere [42,45], the limitations of the present study include those invariably encountered when conducting a retrospective investigation, which is bound by the parameters and precision of the available data. Results for metrics tracked specifically as part of the three-year CCP (e.g., feline intake and euthanasia, and sterilization surgeries) were calculated for 12-month periods that correspond to the beginning (April) and ending (March) months of the program; results customarily tracked by AAWD, apart from the CCP, were based upon the calendar year. For that reason, changes over time for some metrics are reported by calendar year and some by program year; applicable criteria used to calculate changes in metrics are noted herein. Moreover, after initiation of the CCP, some metrics were tracked by both AAWD and BFAS. In order to ensure consistency, data were sourced from BFAS whenever possible; results were obtained directly from AAWD when information from BFAS was unavailable (i.e., years prior to 2011). It was found that whenever common sets of data from each source existed, discrepancies were few and minor (<0.5%). RFCM colony sites were not specifically tracked as part of the CCP; only data indicating the number of cats enrolled by year as part of either the RTF or targeted TNR components of the program were available for analysis. Therefore, for the purposes of this study, RFCM colony sites were defined as those where RTF and targeted TNR activity both occurred during the same calendar year.

Feline intake and euthanasia at AAWD were not formally tracked by zip code; therefore, an assessment of the impact of targeted TNR on these metrics for specific zip codes, as has been formulated elsewhere [10], was not attempted. An assessment of the impact of targeted TNR on feline intake in targeted versus non-targeted zip codes was made, and presented above, as part of a program conducted by AHNM in 2011 and 2012. At the conclusion of the present study, the AHNM-targeted TNR program was ongoing; however, attempts to obtain more recent feline intake data from AHNM were unsuccessful.

Colonies of free-roaming cats were enrolled in the CCP as they were discovered. Colony information, including location and the surgery records of individual cats, was entered into an internal BFAS database. Colony information was updated throughout the program as cats were trapped, sterilized and returned; however, records pertaining to the number of cats at each colony site upon entry into the CCP are incomplete [34]. Consequently, an assessment of the changes in colony size over the course of the program was not possible. In addition, welfare outcomes for cats returned to colony sites were not specifically recorded, precluding analysis. The fates of cats returned to colony sites as part of RTF programs likely warrants further investigation.

## 6. Conclusions

The considerable decline in feline euthanasia at AAWD appears to be associated clearly with implementation of the RTF program. The concurrent reduction in feline intake, including kitten intake, is likely associated with the addition of targeted TNR, including utilization of the RFCM, upon initiation of the CCP. The fact that total feline intake, and that of newborn kittens, declined at AAWD for five consecutive years (2012 to 2016) after an integrated program of RTF and targeted TNR was implemented is notable, and when considered in conjunction with the corresponding decrease in calls to the city about dead cats, likely indicates a reduction in the total population of free-roaming cats present in the Albuquerque community. Additional research focusing on the number of free-roaming cats living in the city of Albuquerque and Bernalillo County as a whole is necessary to confirm this hypothesis.

## Figures and Tables

**Figure 1 animals-08-00055-f001:**
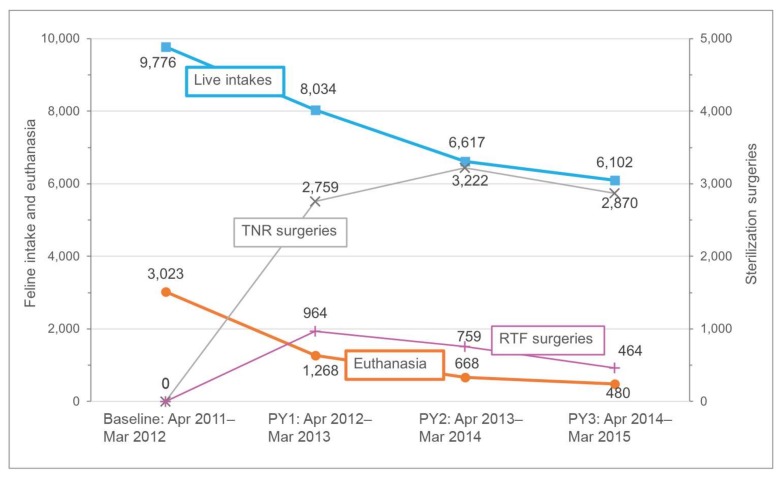
Feline intake, euthanasia, and surgeries: Albuquerque community cat program (CCP).

**Figure 2 animals-08-00055-f002:**
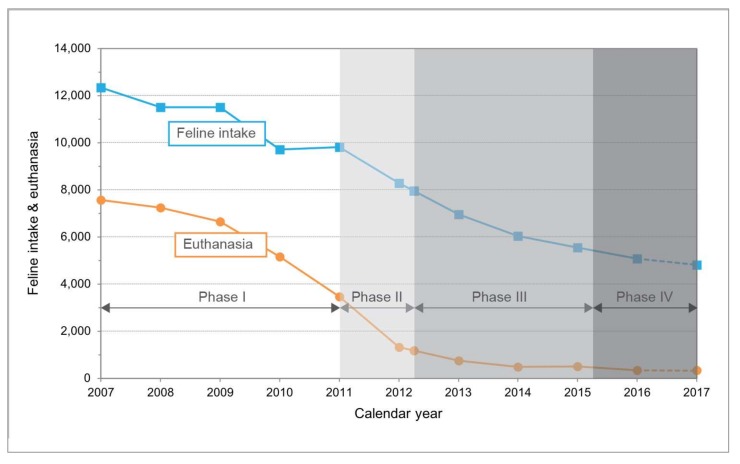
Annual feline intake and euthanasia at Albuquerque Animal Welfare Department, 2007–2017.

**Figure 3 animals-08-00055-f003:**
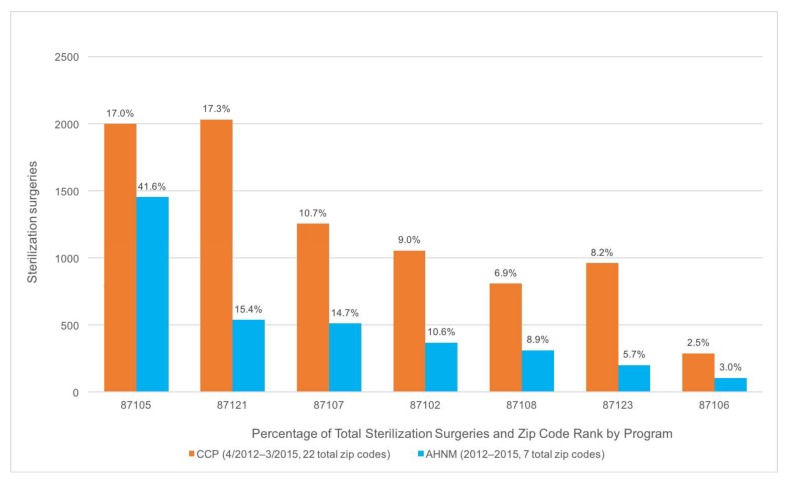
Sterilization surgeries in zip codes targeted by CCP and Animal Humane New Mexico (AHNM).

**Figure 4 animals-08-00055-f004:**
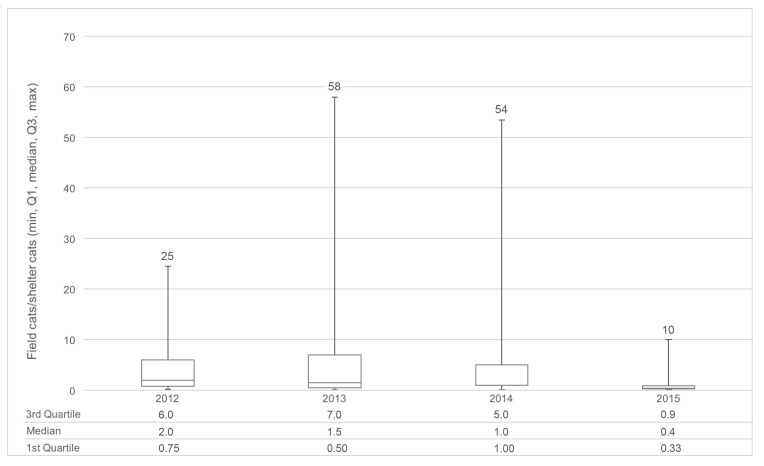
Descriptive statistics (minimum, maximum, median, first and third quartiles) illustrating the impact of “Red-Flag Cat Model” for each program year. For each cat originating as a “shelter intake,” additional “field-origin” cats from the same location were often trapped, sterilized, vaccinated, and returned.

**Table 1 animals-08-00055-t001:** Disposition of cats in the Albuquerque CCP.

Mode of Disposition	Total Cats	Percentage
Returned to Colony	10,738	91.42
Adopted/Rescue	946	8.05
Died	34	0.29
Euthanized	20	0.17
Relocated	6	0.05
Other	2	0.02
Totals	11,746	100

**Table 2 animals-08-00055-t002:** Common shelter metrics before and after implementation of Albuquerque’s CCP.

Shelter Metric	Age Classification	Before Program	After Program
Overall feline intake *		9776	6102
	Kittens **	4441	2468
	<2 months old **	2803	1672
Overall feline euthanasia *		3023	480
	Kittens **	1462	149
Feline euthanasia rate *		30.9%	7.9%
	Kittens **	32.9%	6.0%
Feline LRR **		60.6%	89.5%
Adoptions **		4264	3333
Adoptions/feline intake **		43.5%	60.2%
RTO (returned to owner) **		297	277
RTO/feline intake **		3.0%	5.0%
Calls for dead cat pick-up **		2220	1689

LRR = total live outcomes/total live intake. * Tracked by program year: before program (April 2011 through March 2012); after program (April 2014 through March 2015. ** Tracked by calendar year: before program (year-end 2011); after program (year-end 2015).

**Table 3 animals-08-00055-t003:** Impact of Albuquerque CCP in terms of intake and euthanasia per 1000 human residents, and comparison to similar programs in other communities.

Category and Period of Interest	Community		
	Albuquerque/Bernalillo County, NM	San José, CA	Alachua County, FL (zip code 32601)
Type of program(s)	Targeted TNR & RTF	RTF	Targeted TNR
Program duration	3 years	4 years	2 years
Feline intake/1000 human population			
Before program	14.7	10.2	13.0
After program	8.2	7.0	4.0
Reduction	44.2%	31.4%	69.2%
Euthanasia/1000 human population			
Before program	5.2	7.2	8.0
After program	0.7	1.6	0.4
Reduction	86.5%	77.8%	95.0%
Average sterilizations annually/1000 human population	5.4	2.7	60.5
Source of data	This study	[5]	[10]

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
