# Peer review of "The Impact of an Integrated Program of Return-to-Field and Targeted Trap-Neuter-Return on Feline Intake and Euthanasia at a Municipal Animal Shelter"

_animals, 2018, doi:10.3390/ani8040055_

Round 1

Reviewer 1 Report

This article describes a retrospective study that will be of great interest to individuals who work with animal shelters or humane organizations and to anyone with an interest in overpopulation of cats.

The article is well written and presents valuable information to those who wish to decrease cat intake and euthanasia in local animal shelters.

It is difficult to interpret Figure 5.  Please provide a more complete description of Figure 5 and the significance of the information presented in Figure 5. 

Reviewer 2 Report

The study represents an addition to the growing data regarding the impacts of community cat management programs on shelter intake.  There are a number of issues that need to be addressed prior to publication:

·      The study involves a municipal shelter program, but the Introduction only considers national shelter trends. Statewide (e.g., OH and CO) and other municipal (e.g., Jacksonville and Denver) shelter studies should be included as they are more relevant to this study.

·      Much of the Materials and Methods is neither (e.g., 2.1), and should be moved to the Introduction section.

·      How the programs are funded is irrelevant to the study.  Lines 126 135 should be removed or moved elsewhere.

·      There is no data analysis included in the Materials and Methods. A detailed description of how the results were generated should be added.

·      Figure 2 should be presented as a table, not as a pie chart.

·      It appears that only two data points are used to calculate the changes in the various metrics across the entire study period – and to a decimal point. Trends in the data across all of the years should be analyzed by linear regression to account for natural variation from year to year. The changes should then be calculated from the trendlines, not from the first and last years’ absolute data. Alternatively, error bars calculated from the other years’ data should be added to the reported findings.

·      Lines 229-233 are not a Result and should be moved to the discussion of the timeline of the program.

·      The Discussion or Limitations sections (or both) should include a discussion that the study only identified a correlation between the cat programs and changes in the shelter metrics across time. Therefore, it is not possible to conclude that the programs are the primary cause of the changes.  Claim to causation would require a different study design that includes a strictly defined and monitored control community or ceasing the cat programs and seeing if the shelter metrics start to increase again. While the latter is poor animal welfare practice, the strong claims of causation without proper controls is poor science.  Please point this issue of correlation versus causation out more explicitly in your discussion.

Round 2

Reviewer 1 Report

All concerns I expressed with the original draft of this article have been adequately addressed in the revision. 

Author Response

Thank you for your insights.

Reviewer 2 Report

No further comments

Author Response

Thank you for your insights.